# Health management information system (HMIS) data verification: A case study in four districts in Rwanda

Alphonse Nshimyiryo[1]*, Catherine M. Kirk[1], Sara M. Sauer[2],
Emmanuel Ntawuyirusha[3], Andrew Muhire[3], Felix Sayinzoga[4], Bethany Hedt-Gauthier[1,2,5]

**1** Maternal and Child Health Program, Partners In Health/Inshuti Mu Buzima, Kigali, Rwanda, **2** Department of Biostatistics, Harvard T.H. Chan School of Public Health, Boston, MA, United States of America, **3** Planning, Health Financing and Information Systems, Ministry of Health, Kigali, Rwanda, **4** Maternal, Child and Community Health Division, Rwanda Biomedical Center, Kigali, Rwanda, **5** Department of Global Health and Social Medicine, Harvard Medical School, Boston, MA, United States of America

* anshimyiryo@pih.org

## Abstract

### Introduction

Reliable Health Management and Information System (HMIS) data can be used with minimal cost to identify areas for improvement and to measure impact of healthcare delivery. However, variable HMIS data quality in low- and middle-income countries limits its value in monitoring, evaluation and research. We aimed to review the quality of Rwandan HMIS data for maternal and newborn health (MNH) based on consistency of HMIS reports with facility source documents.

### Methods

We conducted a cross-sectional study in 76 health facilities (HFs) in four Rwandan districts. For 14 MNH data elements, we compared HMIS data to facility register data recounted by study staff for a three-month period in 2017. A HF was excluded from a specific comparison if the service was not offered, source documents were unavailable or at least one HMIS report was missing for the study period. World Health Organization guidelines on HMIS data verification were used: a verification factor (VF) was defined as the ratio of register over HMIS data. A VF<0.90 or VF>1.10 indicated over- and under-reporting in HMIS, respectively.

### Results

High proportions of HFs achieved acceptable VFs for data on the number of deliveries (98.7%;75/76), antenatal care (ANC1) new registrants (95.7%;66/69), live births (94.7%;72/76), and newborns who received first postnatal care within 24 hours (81.5%;53/65). This was slightly lower for the number of women who received iron/folic acid (78.3%;47/60) and tested for syphilis in ANC1 (67.6%;45/68) and was the lowest for the number of women with ANC1 standard visit (25.0%;17/68) and fourth standard visit (ANC4) (17.4%;12/69). The

**Funding:** The author (s) received no specific funding for this work. This study was implemented as part of the All Babies Count (ABC) intervention scale-up in Rwanda between 2017 and 2019, and data collection activities were funded by Grand Challenges Canada Saving Lives at Birth.

**Competing interests:** The authors have declared that no competing interests exist.

majority of HFs over-reported on ANC4 (76.8%;53/69) and ANC1 (64.7%;44/68) standard visits.

## Conclusion

There was variable HMIS data quality by data element, with some indicators with high quality and also consistency in reporting trends across districts. Over-reporting was observed for ANC-related data requiring more complex calculations, i.e., knowledge of gestational age, scheduling to determine ANC standard visits, as well as quality indicators in ANC. Ongoing data quality assessments and training to address gaps could help improve HMIS data quality.

## Introduction

National health management information systems (HMIS) have been established in many low- and middle-income countries (LMICs) for routine collection and management of facility-based data on health care service delivery [1,2]. When the data are of good quality, they can be used–with little-to-no costs–to identify areas that need improvement, to evaluate various health interventions, to inform evidence-based health policies, and to design programs and allocate resources at all levels of the health system [3–5]. However there is evidence of variable quality of HMIS data [6–13], limiting its value in monitoring, evaluation and research in LMICs.

Regular data quality assessment is one of the strategies that can be used for improving the quality of HMIS data in LMICs [14,15]. The World Health Organization (WHO) provides guidelines on data quality review (DQR) through a desk review [16] of data which was previously reported to the HMIS and the verification of HMIS data quality through facility survey [17]. The desk review assesses HMIS data quality in four dimensions: 1) completeness and timeliness of data, 2) internal consistency of reported data, 3) external consistency; and 4) external comparisons of population data. Detailed information on definitions and requirements for the calculation of these dimensions, as well as the application of HMIS data quality assessment through the desk review can be reviewed elsewhere [13,16,18]. The WHO toolkit for data quality review defines a verification factor (VF) as the ratio of recounted data from facility source documents over HMIS data [17]. The application of the VF has been more limited and using variable definitions. However, there is evidence that the level of agreement between HMIS data and records in facility source documents can vary depending on the type of data being collected and be rooted in early stages of collecting those data from facility source documents [11,19–24]. Over- or under-reporting in HMIS data can result from human errors that occur when counting events from source documents or simply not including all events for the reporting period–by omitting some of the necessary source documents or not covering the entire reporting period [11,24]. In addition, intentional over- or under-reporting in HMIS data at the facility level can be motivated by the pressure to meet national targets, whereas, inaccuracies in transferring data from facility source documents to the electronic database can also be associated with excessive workload of staff combined with the pressure to meet reporting deadlines [23].

The Rwanda HMIS was established in 1998 and since 2012, with the goal to improve the quality of routinely-collected health data from community health workers (CHWs) and all HFs across the country, the Rwanda Ministry of Health (MoH) upgraded the HMIS to a web-

based system known as the District Health Information System version 2 (DHIS2) [25]. Recent assessments of the Rwanda HMIS data quality using WHO guidelines have been limited to a desk review of available data in the HMIS [13,26]. These assessments revealed that the quality of the Rwanda HMIS data is high, regarding completeness and internal consistency of reported data for studied HMIS indicators [26]. However, findings from two small-scale studies that tried to compare HMIS data and records in source documents, using a reporting accuracy definition different to the WHO data verification definition, suggest a variable level of agreement between HMIS data and records in source documents [27,28]. One of these two studies that defined accuracy of reporting in HMIS data as a deviation less than or equal to 5% between HMIS and facility source documents data on family planning, antenatal care and delivery, found the overall accuracy of reporting at 70.6% for 37 HFs sampled in three districts in the Eastern Province of Rwanda [28].

Good quality HMIS data would play an important role in identifying areas that need improvement and monitoring progress and evaluating interventions that are designed to address those gaps [29–31]. Rwanda experienced a remarkable progress in reducing mortality among children aged under-five in the past two decades, however the reduction in neonatal deaths (those occurring within 28 days of birth and mainly in health facilities) has been slow [32]. There is no doubt that good quality HMIS data on maternal and newborn healthcare are needed to identify gaps in the existing facility-based care for designing appropriate interventions and for monitoring progress [33]. Since 2013, the "All Babies Count (ABC)" intervention has been implemented to accelerate the reduction in preventable neonatal deaths in Rwanda [34]. The ABC intervention focuses on improved coverage and quality of antenatal, maternity and postnatal care services. ABC is implemented through a joint collaboration of Partners In Health/Inshuti Mu Buzima (PIH/IMB), an international non-profit organization and the Rwanda MoH. However, the evaluation of the impact of these programs has been costly, with a parallel collection of data on program indicators through HMIS and review of facility source documents, given the concerns about poor quality of HMIS data [34]. Therefore, this study uses the WHO guidelines for HMIS data verification to provide evidence on the level of agreement between Rwanda HMIS data and records in source documents of reporting HFs. We calculate VFs for fourteen HMIS data elements that were identified jointly by PIH/IMB and MoH as priority indicators for quality improvement in maternal and newborn health care for 76 HFs (7 hospitals and 69 health centers) that received the ABC intervention between 2017 and 2019 in four districts of Rwanda. The criteria for indicator selection included having clinical relevance to neonatal survival, government priority indicators, and/or being indicators within the WHO standards for improving quality in maternal and newborn care in health facilities [35].

## Methods

### Study design

This was a cross-sectional study that was conducted to assess the quality of Rwanda HMIS data, measured as agreement between HMIS and facility source documents data, for 76 HFs on 14 HMIS data elements related to maternal and newborn health data that were used to monitor quality and progress and to inform quality improvement efforts through the ABC intervention (Table 1). The antenatal care (ANC) register was the source document for 5 data elements that were only reported by health centers (HCs), while the maternity and postnatal care (PNC) registers were source documents for 7 documents that were reported by both HCs and hospitals. Two data elements related to neonatal admissions and deaths were recounted from the neonatology care unit (NCU) register and were only reported by hospitals.

**Table 1. Description of data elements and source of data.**

| Data element | Description | Source of data |
|---|---|---|
| **Data elements only reported by health centers** | | |
| ANC1 | Number of antenatal care (ANC) new registrants | ANC register |
| ANC1_standard | Number of women with ANC1 standard visit (defined in Rwanda as the first ANC visit at <16 weeks' gestation) | ANC register |
| Syphilis_test | Number of women tested for syphilis at ANC1 | ANC register |
| Iron/FA | Number of women received iron/folic acid at ANC1 | ANC register |
| ANC4_standard | Number of women with a 4th ANC standard visit | ANC register |
| **Data elements reported by both health centers and hospitals** | | |
| Deliveries | Number of deliveries in health facilities | Maternity register |
| Live_Births | Number of live births | Maternity register |
| PNC1_Newborn | Number of newborns who received postnatal care (PNC) visit 1 within 24 hours of birth | PNC register |
| **Data elements with rare events reported by both health centers and hospitals** | | |
| Stillbirths | Number of stillbirths (fresh and macerated) | Maternity register |
| Didn't_Cry | Number of live newborns who didn't cry at birth | Maternity register |
| Resusc_Succ | Number of live newborns who didn't cry at birth and were resuscitated successfully | Maternity register |
| LBW | Number of low birth weight babies | Maternity register |
| **Data elements only reported by hospitals** | | |
| Neo_Admissions | Number of neonatal admissions in the hospital neonatology care unit (NCU) | NCU register |
| Neo_Deaths | Number of neonatal deaths in the hospital neonatology care unit (NCU) | NCU register |

## Study setting

This study included 48 HFs in Gakenke and Rulindo districts in Northern Rwanda and 28 HFs in Gisagara and Rusizi districts in Southern and Western Rwanda, respectively. The 76 HFs were grouped into seven hospital catchment areas (HCAs): Nemba District Hospital (15 HFs), Ruli District Hospital (10 HFs), Kinihira Provinical Hospital (9 HFs), Rutongo District Hospital (14 HFs), Gakoma District Hospital (6 HFs), Kibilizi District Hospital (10 HFs) and Mibilizi District Hospital (12 HFs). These MoH-operated facilities were included because they received the ABC intervention, and represented 14% (69/499) of health centers and 15% (7/48) of the hospitals in all 30 districts in Rwanda [36].

The ABC project was originally implemented in 2013 by Partners in Health/Inshuti Mu Buzima (PIH/IMB) in Kayonza and Kirehe districts in Eastern Rwanda, in partnership with the Rwanda MoH [34]. Later on, ABC was scaled-up to Gakenke and Rulindo (July 2017) and then to Gisagara and Rusizi (October 2017). The ABC scale-up project used health facility-based data, collected monthly through the Rwanda HMIS, to monitor indicator progress from baseline to the end of the project and to evaluate its impact. In addition, the project underwent the HMIS data verification process by recounting the same data in standardized HF registers for the same data elements and reporting periods. Five data elements related to antenatal care (ANC) services (ANC new registrants, syphilis testing and iron/folic acid distribution at the first ANC visit and number of women with first and fourth ANC standard visits) were only reported by health centers, whereas two data elements on the number of neonatal admissions

and neonatal deaths in the hospital neonatology care unit were specific to hospitals [37]. Data are recorded using standardized registers developed by the MoH and provided to all HFs (see Table 1 for data sources); women attending ANC are recorded at their first ANC visit and provided an ANC card and an ANC number that facilitates continuity of data recording at the individual level for ANC. This study reports data from the baseline period prior to ABC intervention in the 7 HCAs.

## Sources of data

**HMIS data.** The ABC team worked with the MoH national HMIS team to extract HMIS reports data for the study periods. The HMIS data collection starts from the reporting facility, with clinical staff in each care service registering patients/clients and the care provided to them in standardized registers and/or medical files [38,39]. Then, for monthly reporting to HMIS, the facility data manager ensures the distribution of paper HMIS reporting forms to heads of services by the 25[th] of each month. The head of service collects those data that are relevant to their specific service and submits a completed HMIS report for the previous month to the facility data manager by the 3[rd] day of the month following the month of reporting. For timely reporting, the facility data manager should upload all facility data into DHIS 2 by the 5[th] day of every month. Data verification by facility team and corrections in the system are only allowed between the 5[th] and 15[th] of each month. Any request for changes on the data in the system beyond the 15[th] of each month should be submitted to the central MoH, and access is only granted upon strong justification of the request.

**Recounted data from facility source documents.** A specialized team of two trained ABC data collectors went to all HFs under study and recounted the same data in the standardized HF source documents for the same reporting periods. ABC baseline data—April-June 2017 for 48 HFs in Gakenke and Rulindo districts and July-September 2017 for 28 HFs in Gisagara and Rusizi—were collected during the periods July 31-September 19, 2017 and November 14, 2017-January 11, 2018, respectively. In particular, due to observed variable unit of recording of gestational age (GA) in weeks or months in the ANC register by facility and care provider, ABC data collectors worked with midwives or nurses responsible for providing ANC to standardize the calculation of GA in weeks before recounting data on ANC1 and ANC4 standard visits, as the reporting to HMIS on these data elements is based on GA calculated in weeks. The data collection team used a pregnancy wheel and the data recorded on date of last menstrual period and dates of ANC visits for individual women who attended ANC to determine GA at each visit. For all data elements, data collection was done in consultation with the health facility staff responsible for routine reporting of data into HMIS.

## Data analysis and presentation

We used the WHO DQR guidelines on data verification and system assessment to calculate verification factors (VFs) for each data element [17]. A VF was defined as the ratio of register data to HMIS data (Eq 1). ABC baseline data were aggregated for the three-month reporting period. HMIS and facility source documents data were compared by data element and HF. At the HCA level, a VF was calculated by summing all the non-missing values for each data element and all the reporting HFs under that HCA during the study period. Then, a HCA-level VF was calculated as a ratio of the aggregated recounted data to HMIS data. In addition, a VF for data elements with rare events was only calculated at the HCA level, where aggregated data were compared to avoid denominators with true zero values that would be expected if these data were compared at the HF level. For each data element, we excluded from our analyses any

HFs that were not eligible for reporting on that data element or that had either incomplete HMIS data or missing source documents' data for any month during the reporting period.

$$\text{VF} = \frac{recounted\ number\ of\ events\ from\ facility\ source\ documents}{Reported\ number\ of\ events\ from\ the\ HMIS} \qquad \text{Eq (1)}$$

A VF of 1.00 indicated a perfect match between recounted data and HMIS data. The acceptable margin of error for the discrepancy between HMIS reports data and recounted data in facility registers was (0.90≤VF≤1.10), based on the WHO DQR guidelines. A VF<0.90 or VF>1.10 indicated over-reporting and under-reporting in HMIS data, respectively. We used Stata v.15.1 (Stata Corp, College Station, TX, USA) and R Language and Environment for Statistical Computing for analysis and visual presentation of data [40].

## Ethics

The ABC scale-up project received approval from the Rwanda MoH to access HMIS data for the project's indicators for all HFs that received the intervention. This study was approved by the Rwanda National Ethics Committee approval (Kigali, Rwanda, protocol #0067/RNEC/2017). As this study was completed using de-identified routinely-collected aggregate data, informed consent was not required.

## Results

The proportion of HFs with all data sources by data element for the three-month period was the lowest for iron/folic (87.0%; 60/69), while the proportion of facilities with complete reporting in HMIS was between 90.8% and 100%, and was the lowest for the first postnatal care (PNC1) (85.5%; 65/76) (Table 2). A facility level HMIS data VF was calculated for eight data elements (Table 2 and Fig 1). A high proportion of HFs achieved acceptable VFs for the following data elements: the number of deliveries (98.7%; 75/76), antenatal care (ANC1) new registrants (95.7%; 66/69), live births (94.7%; 72/76), and newborns who received PNC1 visit within 24 hours (81.5%; 53/65). The median VF was 1.00 (interquartile range [IQR]: 0.99–1.00) for HMIS data on deliveries, 1.00 (IQR: 1.00–1.00) for ANC1 data, 1.00 (IQR: 0.99–1.00) for HMIS data on live births and 1.00 (IQR: 0.97–1.02) for PNC1 data.

The proportion of HFs with acceptable VFs was lower for the number of women who received iron/folic acid (78.3%; 47/60) and the number of women who were tested for syphilis on ANC1 (67.6%; 46/68). The median VF was 1.00 (IQR: 0.99–1.00) for iron/folic acid distribution and 1.00 (0.89–1.00) for syphilis testing, respectively. For HFs with a VF out of the acceptable range, 9 in 13 (69.2%) over-reported on iron/folic acid distribution, while 18 in 22 (81.8%) HFs over-reported on HMIS data for syphilis testing on ANC1.

The indicators for which the lowest proportion of HFs obtained acceptable VFs were the number of women with an ANC1 standard visit (at less than 16 weeks' gestational age (GA)) (25.0%; 17/68) and the number of women who completed 4 standard ANC visits (ANC4 standard: ANC1 at less than 16 weeks GA, ANC2 at between 24–28 weeks, ANC3 at between 28–32 weeks and ANC4 at between 36–38 weeks) (17.4%; 12/69). The median VF for HMIS data on ANC1 standard visit was 0.85 (IQR: 0.67–0.99) and 0.75 (IQR: 0.56–0.88) for ANC4 standard visit data. Of the majority of HFs that over- or under-reported on ANC1 standard (75.0%; 51/68) and ANC4 standard (82.6%; 57/69), over-reporting was at 93.0% (53/57) and 86.3% (44/51) for ANC4 and ANC1 standard visits.

When data were aggregated at the HCA level (Table 3), all 7 HCAs achieved acceptable VFs for HMIS data on the number of ANC1, deliveries, live births and newborns who received PNC1 within 24 hours of birth. Five and four in seven HCAs achieved acceptable VFs for

**Table 2. Facility-level verification factors (VF) by data element.**

| Data element name (n)[a] | Facilities that provide each service and with complete source documents' data, n (%) | Facilities that provide each service and with complete HMIS data, n (%) | Facilities for which source data exactly match HMIS data (VF = 1) n (%) | Facilities with an acceptable margin of error for discrepancies between HMIS and facility source documents data 0.90≤VF≤1.10 n (%) | Facilities that over-reported by more than 10% (VF <0.90) n (%) | Facilities that under-reported by more than 10% (VF >1.10) n (%) | Median facility-level VF Median [IQR] |
|---|---|---|---|---|---|---|---|
| ANC1[b] (n = 69) | 69 (100.0) | 69 (100.0) | 49 (71.0) | 66 (95.7) | 3 (4.3) | 0 (0.0) | 1.00 [1.00–1.00] |
| ANC1_standard[c] (n = 68) | 69 (100.0) | 68 (98.6) | 5 (7.3) | 17 (25.0) | 44 (64.7) | 7 (10.3) | 0.85 [0.67–0.99] |
| Syphilis_test[d] (n = 68) | 68 (98.6) | 69 (100.0) | 25 (36.8) | 46 (67.6) | 18 (26.5) | 4 (5.9) | 1.00 [0.89–1.00] |
| Iron/FA[e] (n = 60) | 60 (87.0) | 69 (100.0) | 29 (48.3) | 47 (78.3) | 9 (15.0) | 4 (6.7) | 1.00 [0.99–1.00] |
| ANC4_standard[f] (n = 69) | 69 (100.0) | 69 (100.0) | 2 (2.9) | 12 (17.4) | 53 (76.8) | 4 (5.8) | 0.75 [0.56–0.88] |
| Deliveries[g] (n = 76) | 76 (100.0) | 76 (100.0) | 42 (55.3) | 75 (98.7) | 1 (1.3) | 0 (0.0) | 1.00 [0.99–1.00] |
| Live_Births[h] (n = 76) | 76 (100.0) | 76 (100.0) | 39 (51.3) | 72 (94.7) | 4 (5.3) | 0 (0.0) | 1.00 [0.99–1.00] |
| PNC1_Newborn[i] (n = 65) | 71 (93.4) | 69 (90.8) | 10 (15.4) | 53 (81.5) | 5 (7.7) | 7 (10.8) | 1.00 [0.97–1.02] |

[a] Final number of reporting facilities with both complete source documents and HMIS data; antenatal care (ANC)-related services are only provided at the health center, so reports were complete if n = 69 health centers reported and n = 76 for services which are provided at both health centers and hospitals

[b] Number of antenatal care (ANC) new registrants

[c] Number of women with first ANC standard visit

[d] Number of women tested for syphilis at first ANC visit

[e] Number of women who received iron/folic acid at first ANC visit

[f] Number of women with a fourth ANC standard visit

[g] Number of deliveries in health facilities

[h] Number of live births; and

[i] Number of newborns who received postnatal care (PNC) visit 1 within 24 hours of birth

HMIS data on the number of pregnant women who received iron/folic acid and were tested for syphilis on ANC1, respectively. Two in seven HCAs achieved acceptable VFs for HMIS data on the number of women with ANC1 standard visit. With a VF ranging between 0.59 and 0.86, none of the HCAs obtained an acceptable VF, and all HCAs over-reported on HMIS data, for the number of women with ANC4 standard visit.

A VF was calculated at the HCA level only for the six HMIS data elements with rare events or for data elements concerning a service that was only provided at the hospital level (Table 4). Six in seven hospitals achieved acceptable VFs for the number of admissions in the hospital NCU and 5 in 7 hospitals obtained an acceptable VF for the number of neonatal deaths in NCU. The majority (5/7) of sites also achieved acceptable VFs for the number of stillbirths and the number of babies with low weight at birth. Less than half (3/7) of sites obtained acceptable VFs for the number of live newborns who needed resuscitation and were resuscitated successfully. Only 2 in 7 sites achieved an acceptable VF for the number of live newborns who didn't cry at birth and the majority (4/7) of sites under-reported on this data element.

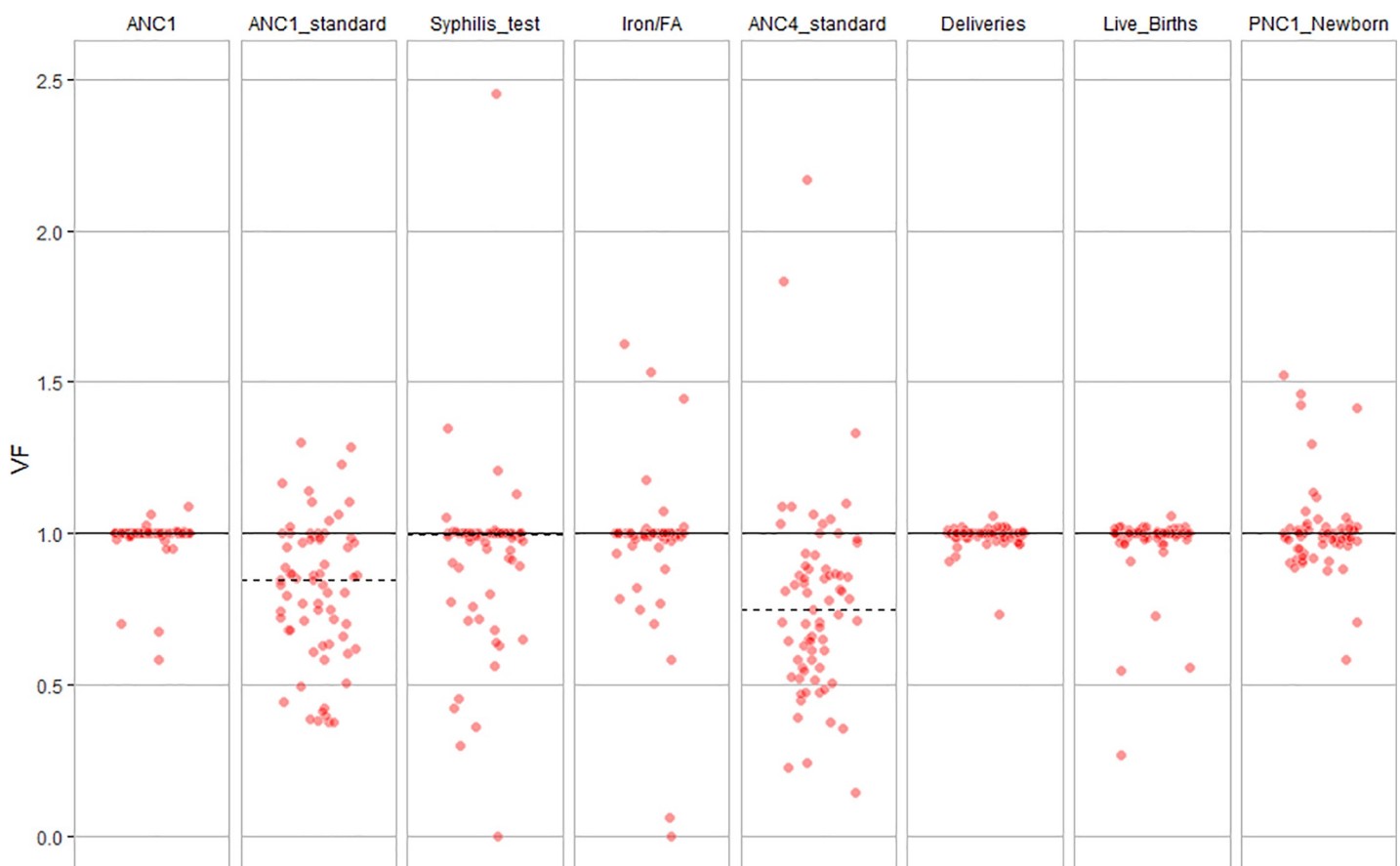

**Fig 1. Facility-level verification factors (VF) by data element.** The median is not visible when it is very close to or equal to 1.

## Discussion

In this study, we assessed the quality of the Rwanda HMIS data, measured as the level of agreement between HMIS data and records in facility source documents, using data from 76 public

**Table 3. Hospital catchment-level verification factor (VF) for aggregated data by data element, a VF>1.10 or <0.90 is highlighted.**

| Data element | Nemba DH* | Ruli DH | Kinihira PH** | Rutongo DH | Gakoma DH | Kibilizi DH | Mibilizi DH |
|---|---|---|---|---|---|---|---|
| Number of antenatal care (ANC) new registrants | 0.97 | 1.01 | 0.99 | 1.00 | 1.00 | 0.93 | 0.98 |
| Number of women with first ANC standard visit | **0.82** | **0.82** | 0.90 | 0.93 | **0.80** | **0.51** | **0.72** |
| Number of women tested for syphilis at first ANC visit | **0.79** | **0.65** | 1.03 | 0.96 | 0.95 | **0.86** | 1.10 |
| Number of women who received iron/folic acid at first ANC visit | 0.95 | 1.01 | 0.99 | **0.81** | 0.99 | 0.95 | **0.89** |
| Number of women with a fourth ANC standard visit | **0.68** | **0.83** | **0.73** | **0.86** | **0.82** | **0.59** | **0.69** |
| Number of deliveries in health facilities | 1.00 | 1.00 | 1.00 | 1.00 | 1.00 | 0.95 | 1.00 |
| Number of live births | 1.00 | 0.99 | 1.00 | 1.00 | 0.95 | 0.95 | 0.99 |
| Number of newborns who received postnatal care (PNC) visit 1 within 24 hours of birth | 1.00 | 0.99 | 1.00 | 0.98 | 1.03 | 0.91 | 0.99 |

*DH: District Hospital

**PH: Provincial Hospital

**Table 4. Hospital catchment-level verification factors (VF) for data elements with rare events, a VF>1.10 or <0.90 is highlighted.**

| Data element | Nemba DH* | Ruli DH | Kinihira PH** | Rutongo DH | Gakoma DH | Kibilizi DH | Mibilizi DH |
|---|---|---|---|---|---|---|---|
| Number of stillbirths (fresh and macerated) | **0.89** | 1.00 | **0.88** | 1.00 | 1.04 | 1.06 | 1.00 |
| Number of live newborns who didn't cry at birth | **1.19** | **1.49** | **1.17** | **1.16** | 0.93 | **1.10** | **0.77** |
| Number of live newborns who didn't cry at birth and were resuscitated successfully | 0.96 | **1.48** | 1.04 | **1.20** | **1.19** | 1.06 | **0.67** |
| Number of low birth weight babies | 1.10 | 1.00 | **1.23** | 1.07 | 1.10 | **1.20** | 0.93 |
| Number of neonatal admissions in the NCU | 1.00 | 1.00 | 1.00 | 0.92 | 1.00 | 1.00 | **0.88** |
| Number of neonatal deaths in the NCU | **1.22** | 1.00 | 1.07 | 1.08 | 1.00 | 0.96 | **1.38** |

*DH: District Hospital

**PH: Provincial Hospital

HFs, in Northern, Southern and Western Rwanda. Fourteen HMIS data elements were selected for this study, considering their importance in identifying gaps and monitoring progress towards the improvement of maternal and newborn health for reducing preventable neonatal deaths in Rwanda. Our findings suggested several strengths while also observing variation in the quality of HMIS data by type of data element, which is consistent with other HMIS DQA studies in Rwanda and other Sub-Saharan African countries [27,41].

Notably, this verification showed high level of agreement between data reported to HMIS and records in facility source documents for the number of ANC1, deliveries and live births. These data elements are among the WHO recommended core indicators for DQR in HMIS data on maternal health [17]. The quality of Rwanda HMIS data on these data elements is higher than what was found in HMIS data verifications for the same data in Ethiopia [41] and Nigeria [42] and similar for the ANC1 indicator in Malawi [22]. This verification of the Rwanda HMIS data also showed similar patterns of data quality of HMIS data elements at the facility level and when HFs were grouped by HCA. This finding may indicate that there were common challenges for accurate reporting to HMIS, regardless of the HF geographical location. This is a different finding to that found by other studies that revealed quality of routinely collected health data in Africa varied by geographical location of reporting HFs [18,43]. Consistent level of HMIS data quality across Rwanda may be the result of adherence of HFs to the Rwanda MoH's existing standard operating procedures for high quality HMIS data [25] and performance based financing system that includes regular review of facility records with specific focus on maternal and child health [44].

However, there was poor quality of HMIS data on the number of ANC1 and ANC4 standard visits with a general trend of over-reporting. A systematic review of immunization data quality identified insufficient human resources and limited healthcare worker capacity for reporting and using data as key issues that contribute to poor data quality [23]. The accuracy of reporting on these data elements might be dependent on the health care provider's knowledge of how to calculate the pregnancy's gestational age in weeks and how to correctly schedule the ANC standard visits, as well as the availability of tools, mainly pregnancy wheels, that facilitate these calculations. It was observed by the study data collection team, that data in registers were recorded in ways that were not compatible with HMIS reporting, for example–gestational age at first visit recorded in the register in months and then reported into HMIS based on cutoffs in weeks which may contribute to challenges in reporting incompatible information out of the source documents into HMIS. In addition, an analysis of the ABC baseline data on the availability of essential medical equipment and supplies at facilities that received the ABC intervention, revealed that only 44.9% (31/69) of health centers were having a pregnancy

wheel in the pre-intervention period. The low quality of HMIS data on these data elements might be justified by the findings of a recent study on quality of ANC services provision in HFs conducted in 13 sub-Sahara African countries, including Rwanda [45]. Using data from Service Provision Assessments (SPA) and Service Availability and Readiness Assessments (SARA) surveys, it was revealed that the proportion of HFs providing ANC services with ANC-trained staff was less than half in 6 of the 13 countries. In addition, the median proportion of HFs with ANC guidelines was 62.3%. In Rwanda in particular, only 31.2% of 432 facilities providing ANC services had ANC guidelines and only 79.8% of these facilities had staff trained in ANC [45]. This may particularly affect the observed over-reporting on the number of women who completed four ANC standard visits which requires also knowing the standard visit schedule for ANC to know whether women completed the four visits at the correct time, versus just reporting the number of women who came for four visits at any time.

Further, the over-reporting of key quality of care elements such as iron/folic acid supplementation and syphilis testing in ANC, are concerning. In addition to challenges of correct reporting of ANC coverage identified in this study, these data elements [46] are included to help understand the content and quality of ANC visits that women receive. These are critical interventions for prevention of stillbirths, genetic abnormalities, and poor neonatal outcomes [47,48]. Over-reporting masks an important problem that has major implications for the health of women and newborns.

## Study limitations

First, this study included only 14 Rwanda HMIS data elements related to maternal and newborn care and HFs that were selected to receive the ABC scale-up intervention, and this non-random sample might not be representative of all facilities in Rwanda. However, the fact that these facilities were located in different parts of the country, and that the results from this study show similar variations in data quality by HMIS data element across all geographical locations, we are confident that the findings of this study can help with understanding the level of agreement between HMIS data and facility source documents' records for the considered data elements. We also believe that the findings of this study will be useful to other studies for the verification of more HMIS data elements or on a larger scale in Rwanda. Second, this study only assessed the quality of HMIS data by focusing on the concordance between HMIS and facility source documents data. The true accuracy of the source documents is not known, and is a critical component of data quality that requires further evaluation.

## Conclusions

Findings of this study suggest the variation of HMIS data quality by data element and similar patterns of reporting accuracy, irrespective of the geographical locations of a health facility. Reporting to HMIS was less accurate for some data elements, particularly those that are more complex to generate. This challenge to accurate reporting by HFs has implications for decision-making on key interventions affecting maternal and newborn outcomes. Ongoing regular data quality assessments, promoting the use of HMIS data for quality improvement in health care delivery at the facility level, and training to address gaps could help improve HMIS data to be used in program evaluations.

## Supporting information

**S1 Dataset.**
(DTA)

## Acknowledgments

We acknowledge the contributions of Robert M. Gatsinzi and Ibrahim Hakizimana from the monitoring and evaluation team for data collection in health facility source documents. We are also grateful to the leadership, nurses and midwives of health facilities involved in this study, for their critical support in facilitating data collection for this study.

## Author Contributions

**Conceptualization:** Alphonse Nshimyiryo, Catherine M. Kirk, Sara M. Sauer, Bethany Hedt-Gauthier.

**Data curation:** Alphonse Nshimyiryo, Sara M. Sauer, Emmanuel Ntawuyirusha, Andrew Muhire.

**Formal analysis:** Alphonse Nshimyiryo, Sara M. Sauer.

**Investigation:** Alphonse Nshimyiryo, Catherine M. Kirk, Sara M. Sauer, Emmanuel Ntawuyirusha, Andrew Muhire, Felix Sayinzoga, Bethany Hedt-Gauthier.

**Methodology:** Alphonse Nshimyiryo, Catherine M. Kirk, Sara M. Sauer, Felix Sayinzoga, Bethany Hedt-Gauthier.

**Project administration:** Alphonse Nshimyiryo, Catherine M. Kirk.

**Resources:** Alphonse Nshimyiryo, Catherine M. Kirk, Sara M. Sauer, Emmanuel Ntawuyirusha, Andrew Muhire, Felix Sayinzoga, Bethany Hedt-Gauthier.

**Software:** Alphonse Nshimyiryo.

**Supervision:** Bethany Hedt-Gauthier.

**Validation:** Catherine M. Kirk, Sara M. Sauer, Emmanuel Ntawuyirusha, Andrew Muhire, Felix Sayinzoga, Bethany Hedt-Gauthier.

**Visualization:** Alphonse Nshimyiryo, Sara M. Sauer.

**Writing – original draft:** Alphonse Nshimyiryo, Catherine M. Kirk.

**Writing – review & editing:** Alphonse Nshimyiryo, Catherine M. Kirk, Sara M. Sauer, Emmanuel Ntawuyirusha, Andrew Muhire, Felix Sayinzoga, Bethany Hedt-Gauthier.

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
