## [Decision Letter · Decision Letter 0]

3 Feb 2020

PONE-D-19-35132

Rwanda Health Management Information System (HMIS) data verification: A case of seventy-six health facilities in four districts of Rwanda

PLOS ONE

Dear Mr. Nshimyiryo,

Thank you for submitting your manuscript to PLOS ONE. After careful consideration, we feel that it has merit but does not fully meet PLOS ONE’s publication criteria as it currently stands. Therefore, we invite you to submit a revised version of the manuscript that addresses the points raised during the review process.  Please address the methodology and analysis related concerns raised by the two reviewers in much detail.

We would appreciate receiving your revised manuscript by Mar 19 2020 11:59PM. To enhance the reproducibility of your results, we recommend that if applicable you deposit your laboratory protocols in protocols.io, where a protocol can be assigned its own identifier (DOI) such that it can be cited independently in the future. For instructions see: http://journals.plos.org/plosone/s/submission-guidelines#loc-laboratory-protocols

We look forward to receiving your revised manuscript.

Kind regards,

Rakhi Dandona

Academic Editor

PLOS ONE

Journal Requirements:

Additional Editor Comments (if provided):

1. The methods section needs more detail as highlighted by both the reviewers. Please also provide information on the level and extent of completeness and missingness of the HMIS variables and the data sources.

2. Please provide details of how the verification factor was calculated at the HCA level. It refers to the “hospital catchment area”, but no details are provided about from where the catchment data were sought.

3. Internal consistency between the HMIS variables and data sources can be assessed, and it will add value to this paper.

4. Overall, the study shows reasonable quality of HMIS other than for ANC, which seems a bit unexpected as the assumption was for this to be of poor quality. Discussion does not highlight this positive aspect of the study findings, and focusses mainly on the two ANC variables which were found of a lesser quality.

5. It will also be useful for the authors to comment on the extent of identifying unique pregnant women across the continuum of care within the current HMIS.

Reviewers' comments:

Reviewer's Responses to Questions

**Comments to the Author**

1. Is the manuscript technically sound, and do the data support the conclusions?

Reviewer #1: Partly

Reviewer #2: Partly

2. Has the statistical analysis been performed appropriately and rigorously? 

Reviewer #1: Yes

Reviewer #2: Yes

3. Have the authors made all data underlying the findings in their manuscript fully available?

Reviewer #1: Yes

Reviewer #2: Yes

4. Is the manuscript presented in an intelligible fashion and written in standard English?

Reviewer #1: Yes

Reviewer #2: Yes

5. Review Comments to the Author

Reviewer #1: Overall comments:

This study presents an analysis of the routine data quality dimension of internal consistency for a set of maternal and newborn care data elements, using the WHO verification factor metric to demonstrate the accuracy of facility reporting. This paper is a helpful contribution to the literature for understanding an important aspect of the quality of routine data in HMIS to monitor maternal and neonatal care and extends the evidence of previous studies on completeness and internal consistency for Rwanda. I hope the authors will find the comments helpful as they revise the paper.

Major comments:

Introduction: Overall comment that the rationale and literature review could be strengthened.

Introduction, paragraph 2: Aside from introducing the four data quality dimensions of the WHO data quality review toolkit for facility data, the purpose of the remainder of the paragraph is unclear:

• There is an assertion that the WHO guidelines, including the VF metric, are limited in implementation given that the guidelines have been out for over a decade. Then the next sentence noted 5 studies which used the VF. In the literature, there are more than the five studies cited that used the WHO VF by name, and even more studies that use a percentage or ratio to relate the counts of data in facility records to the reported facility data. Accuracy of reporting along with completeness are the most analysed dimensions of data quality (please refer to systematic reviews on health data quality for both high and low-middle income countries).

• In this paragraph and the next paragraph, there is an emphasis on the WHO verification factor metric definition, 0.90<vf<1.10, way.="">

Introduction, paragraph 2, first sentence “The World Health Organization…coverage rates”: Please note that the four dimensions named here are from the ‘WHO data quality health facility data quality report card’ and not from the ‘WHO data quality review: a toolkit for facility data quality assessment’ which frames the dimensions slightly differently. Please update the dimensions according to your preferred reference.

Methods: Please include a brief description for why these indicators were selected (a sentence or two).

Methods: As this metric assesses the accuracy of facility reporting, please describe how the data is captured, summarized, reported, and subsequently entered into DHIS 2.

Results, paragraph 1, first sentence “The proportion of HFs … iron/folic acid (87%; 60/69)”: This sentence provides important information on completeness of data that is difficult to readily calculate from Table 2. It puts the VF results in context. Consider including a column in Table 2, after the data element, that notes the completeness of the data as a proportion of the facilities that are providing that service.

Results, figure 1: Excellent figure. Consider adding a legend which reminds the reader which directions represent under/overreporting.

Table 4, “rare data elements”: I wouldn’t call these rare data elements as they are regularly reported into HMIS. Perhaps “rare events” or “rare outcomes”?

Discussion: Overall comment that the literature review should be updated to reflect where the current study fits in.

Discussion, paragraph 2, 3rd sentence “These quality of Rwanda HMIS data… same data in other Sub-Sahara African countries”: The sentence notes “countries” but references only one study in one country.

Discussion, paragraph 2, 6th sentence “This is a different finding to that found by other studies…by geographical location of reporting HFs”: Again, the sentence notes “other studies” but references only one study.

Discussion, paragraph 3: Please reflect more on the poor level of agreement, as there are notable directions in the level of agreements based on data element which are not elaborated – which indicators under/overreport and potential reasons.

Discussion, paragraph 3, 2nd sentence “The accuracy of reporting on these data elements might be dependent on the health workers knowledge of how to calculate the pregnancy’s gestational age in weeks and how to correctly schedule the ANC standard visits, as well as the availability of tools, mainly pregnancy wheels, that facilitate these calculations.” While this may be true in terms of the validity of the documentation, the data verification exercise assesses the ability of the health care worker to report as expected based on the source documents. The external research team, whose data collection is being used as the reference standard for the recounted data, does not also measure the women for gestational age, schedule the appointments, etc. They are looking at the same data source, as the health facility staff would, for determining which numbers would be counted as ANC1_standard versus ANC new registrants.

Reviewer #2: This a relevant paper as it addresses a crucial aspect of Health Management Information System (HMIS) data that is data quality assessment. HMIS are promising source data but data quality remains challenging in Sub-Saharan Africa countries including Rwanda. Unlike to most of the studies focusing on desk review of routine health information system data, this study verifies facility source documents and the level of agreement with national records.

However, it is a pity that the study is limited to only 76 districts within 4 districts that are non-representative of the entire Rwanda. In this is regard, it may be valuable to provide a brief context description of Rwanda heath system and health information system including the number of health facilities and districts in the country, the public and private sector, HMIS data collection process and data processing as that may impact the level of agreement.

Suggestion to revise the title that is a bit long and to make it more attractive (e.g. "Health Management Information System (HMIS) data verification: A case study in four districts in Rwanda"

In the abstract, good to clearly present the objective of the study, as this is not obvious, as well as improving the results and conclusion sections.

My main comments are related to the objective of the study, analysis performed and the discussions of the results. I agree that the verification of level of agreement between HMIS data and facility source documents data is a relevant objective. However, I do think that you may be able to go beyond this objective and tackle additional analysis. You may for instance carry out additional analysis like completeness of reporting and internal consistency for both HMIS data and facility source documents in order to highlight the impact of the lack of agreement between both sources on data quality. I wonder whether that is possible based on available data or at this stage, but it may be worthwhile to assess the impact of data accuracy (level of agreement) on data quality (e.g. completeness of reporting or internal consistency). A question can be to know whether districts with good level of agreement report data with better quality. Since you stated that (p.18) "this verification showed high level of agreement between data reported to HMIS and records in facility source documents for the number of ANC1, deliveries and live births", it may be interesting to assess data quality in districts with good agreement versus districts with low agreement.

To assess whether the source of data matter, it may be interesting to provide, even as an appendix, a table presenting the verification factor by source of data (ANC register, maternity register, PNC register, NCU register), and address a bit that in the analysis.

The discussion section needs to be deeply revisited to reflect more the expectations from a classical discussion section, discuss more the discrepancies between both sources, relevant factors (e.g. unmotivated, poor trained or overworked health personnel, disinterest for health data, etc., problem of equipment, potential issues regarding the transfer of data, data entry errors, etc.), implications of the results.

As an explication you stated (p. 18-19) "However, there was poor quality of HMIS data on the number of ANC1 and ANC4 standard visits. The accuracy of reporting on these data elements might be dependent on the health care provider’s knowledge of how to calculate the pregnancy’s gestational age in weeks and how to correctly schedule the ANC standard visits, as well as the availability of tools, mainly pregnancy wheels, that facilitate these calculations". Through this explanation, you are trying to address the reasons related to the true accuracy of the source documents, instead of providing relevant reasons explaining the lack of agreement between HMIS data and facility source documents. Suggestions to provide relevant reasons/explanations and in line with findings.

Row 233; Check the median VF for syphilis as it does not with data in Table 2.

Table 3: Write properly the label of data elements.

 </vf<1.10,>

6. PLOS authors have the option to publish the peer review history of their article (what does this mean?). If published, this will include your full peer review and any attached files.

Reviewer #1: No

Reviewer #2: No

---

## [Author Response · Author response to Decision Letter 0]

19 Mar 2020

Dear Academic Editor of PLOS ONE,

Thank you for the opportunity to revise our manuscript to meet PLOS ONE’s publication criteria. We have addressed each reviewer’s comments and points raised by Editor in this revised version of the manuscript.

Editor’s comments:

1)The methods section needs more detail as highlighted by both the reviewers. Please also provide information on the level and extent of completeness and missingness of the HMIS variables and the data sources.

Thank you. We have now added two columns to Table 2 to report the proportion of health facilities that provide each service and had completed source documents’ data and HMIS data for the reporting period. We also added a note indicating that all reporting health facilities had complete HMIS data.

2) Please provide details of how the verification factor was calculated at the HCA level. It refers to the “hospital catchment area”, but no details are provided about from where the catchment data were sought.

We have now provided more details on the process of calculating a verification factor at the HCA, where for each data element, all non-missing values were summed for all the health facilities under that HCA during the study period. See page 11, line 211-217, which reads as “At the HCA level, a VF was calculated by summing all the non-missing values for each data element and all the reporting HFs under that HCA during the study period. Then, a HCA-level VF was calculated as a ratio of the aggregated recounted data to HMIS data. In addition, a VF for data elements with rare events was only calculated at the HCA level, where aggregated data were compared to avoid denominators with true zero values that would be expected if these data were compared at the HF level”.

3) Internal consistency between the HMIS variables and data sources can be assessed, and it will add value to this paper.

Thank you for this comment. We might have assessed the internal consistency using methods described in the World Health Organization (WHO) data quality assessment guidelines, however a short study period (3 months) was a limitation to do so. The WHO guidelines on HMIS data quality assessment (DQA) recommend a minimum of 12 months’ data in order to assess the internal consistency between related indicators. The number of women with ANC4 standard visit would be expected to be a portion of those who had an ANC1 standard visit in the past and the ratio between them to be 1 or below, but the fact that the ANC4 is a cumulative variable as those who completed the four standard visits during the study period were those who had their first standard ANC1 in six months before. Therefore, as our study period was just 3 months, it’s more likely that we may falsely claim inconsistency or consistency between ANC1 standard and ANC4 standard visits data elements. Using the data for 3 months, the level of consistency between ANC1 standard and ANC4 standard was quite similar for both HMIS and recounted data from source documents. We have shared this here, but not included in the paper due to the limitations noted. 

Hospital catchment area (HCA, n*) Health facilities with internal consistency between indicators (ANC4/ANC1≤1) for recounted data from source documents

n (%) Health facilities with internal consistency between indicators (ANC4/ANC1≤1) for HMIS data

n (%)

Nemba DH (n=13) 11 (84.6) 8 (61.5)

Ruli DH (n=9) 5 (55.6) 7 (77.8)

Kinihira PH (n=8) 5 (62.5) 5 (62.5)

Rutongo DH (n=13) 12 (92.3) 12 (92.3)

Gakoma DH (n=5) 4 (80.0) 4 (80.0)

Kibilizi DH (n=9) 4 (44.4) 5 (55.6)

Mibilizi DH (n=11) 8 (72.7) 7 (63.6)

Overall (n=68) 49 (72.1) 48 (70.6)

Note: n*, is the number of reporting health facilities in each hospital catchment area (HCA)

4) Overall, the study shows reasonable quality of HMIS other than for ANC, which seems a bit unexpected as the assumption was for this to be of poor quality. Discussion does not highlight this positive aspect of the study findings, and focusses mainly on the two ANC variables which were found of a lesser quality.

Thank you for this comment. We have now highlighted the high quality of the Rwanda HMIS data for some data elements related to maternal and newborn health by comparing our findings to what was found by other HMIS data verification studies. This now reads as: “Notably, this verification showed high level of agreement between data reported to HMIS and records in facility source documents for the number of ANC1, deliveries and live births. These data elements are among the WHO recommended core indicators for DQR in HMIS data on maternal health17. The quality of Rwanda HMIS data on these data elements is higher than what was found in HMIS data verifications for the same data in Ethiopia40 and Nigeria41 and similar for the ANC1 indicator in Malawi22.” (see page 21, lines 341-346).

5) It will also be useful for the authors to comment on the extent of identifying unique pregnant women across the continuum of care within the current HMIS.

Thank you. We have added to the “Methods section/Study setting” the information on the extent of identifying unique pregnant women across the continuum of care within the Rwanda HMIS (see page 9, lines 172-175). The text reads as follows: “Data are recorded using standardized registers developed by the MoH and provided to all HFs (see Table 1 for data sources); women attending ANC are recorded at their first ANC visit and provided an ANC card and an ANC number that facilitates continuity of data recording at the individual level for ANC.”

Reviewer #1’s comments: 

Overall comments: This study presents an analysis of the routine data quality dimension of internal consistency for a set of maternal and newborn care data elements, using the WHO verification factor metric to demonstrate the accuracy of facility reporting. This paper is a helpful contribution to the literature for understanding an important aspect of the quality of routine data in HMIS to monitor maternal and neonatal care and extends the evidence of previous studies on completeness and internal consistency for Rwanda. I hope the authors will find the comments helpful as they revise the paper.

Thank you.

Major comments:

1) Introduction: Overall comment that the rationale and literature review could be strengthened.

• Introduction, paragraph 2: Aside from introducing the four data quality dimensions of the WHO data quality review toolkit for facility data, the purpose of the remainder of the paragraph is unclear:

o There is an assertion that the WHO guidelines, including the VF metric, are limited in implementation given that the guidelines have been out for over a decade. Then the next sentence noted 5 studies which used the VF. In the literature, there are more than the five studies cited that used the WHO VF by name, and even more studies that use a percentage or ratio to relate the counts of data in facility records to the reported facility data. Accuracy of reporting along with completeness are the most analysed dimensions of data quality (please refer to systematic reviews on health data quality for both high and low-middle income countries). In this paragraph and the next paragraph, there is an emphasis on the WHO verification factor metric definition, 0.90

Thank you very much for this comment. We have revised the paragraph on pages 4-5 (lines 85-97) and now reads as follows: “The WHO toolkit for data quality review defines a verification factor (VF) as the ratio of recounted data from facility source documents over HMIS data17. The application of the VF has been more limited and using variable definitions. However, there is evidence that the level of agreement between HMIS data and records in facility source documents can vary depending on the type of data being collected and be rooted in early stages of collecting those data from facility source documents11,19–24. Over- or under-reporting in HMIS data can result from human errors that occur when counting events from source documents or simply not including all events for the reporting period – by omitting some of the necessary source documents or not covering the entire reporting period11,24. In addition, intentional over- or under-reporting in HMIS data at the facility level can be motivated by the pressure to meet national targets, whereas, inaccuracies in transferring data from facility source documents to the electronic database can also be associated with excessive workload of staff combined with the pressure to meet reporting deadlines23”. We also included a systematic review paper on HMIS data verification for additional notes on barriers to data quality (Wetherill et al., 2017).

• Introduction, paragraph 2, first sentence “The World Health Organization…coverage rates”: Please note that the four dimensions named here are from the ‘WHO data quality health facility data quality report card’ and not from the ‘WHO data quality review: a toolkit for facility data quality assessment’ which frames the dimensions slightly differently. Please update the dimensions according to your preferred reference.

Thank you. We have now updated the four dimensions for HMIS data quality review through a desk review according to the 2017 World Health Organization guidelines on HMIS data quality review (see page 4, lines 80-83). This reads as: “The desk review assesses HMIS data quality in four dimensions: 1) completeness and timeliness of data, 2) internal consistency of reported data, 3) external consistency; and 4) external comparisons of population data”.

2. Methods: 

• Please include a brief description for why these indicators were selected (a sentence or two).

Thank you. We have clarified in the introduction the reason why this study included 14 HMIS data elements related to maternal and newborn health care. It reads as follows: “We calculate VFs for fourteen HMIS data elements that were identified jointly by PIH/IMB and MoH as priority indicators for quality improvement in maternal and newborn health care for 76 HFs (7 hospitals and 69 health centers) that received the ABC intervention between 2017 and 2019 in four districts of Rwanda. The criteria for indicator selection included having clinical relevance to neonatal survival, government priority indicators, and/or being indicators within the World Health Organization standards for improving quality in maternal and newborn care in health facilities” (see page 6, lines 129-135).

• As this metric assesses the accuracy of facility reporting, please describe how the data is captured, summarized, reported, and subsequently entered into DHIS 2.

Thank you. We have now described the sources of data in details under the methods section (see page 10, lines 180-190), and it reads as follows: “The HMIS data collection starts from the reporting facility, with clinical staff in each care service registering patients/clients and the care provided to them in standardized registers and/or medical files37,38. Then, for monthly reporting to HMIS, the facility data manager ensures the distribution of paper HMIS reporting forms to heads of services by the 25th of each month. The head of service collects those data that are relevant to their specific service and submits a completed HMIS report for the previous month to the facility data manager by the 3rd day of the month following the month of reporting. For timely reporting, the facility data manager should upload all facility data into DHIS 2 by the 5th day of every month. Data verification by facility team and corrections in the system are only allowed between the 5th and 15th of each month. Any request for changes on the data in the system beyond the 15th of each month should be submitted to the central MoH, and access is only granted upon strong justification of the request”.

3. Results:

• Results, paragraph 1, first sentence “The proportion of HFs … iron/folic acid (87%; 60/69)”: This sentence provides important information on completeness of data that is difficult to readily calculate from Table 2. It puts the VF results in context. Consider including a column in Table 2, after the data element, that notes the completeness of the data as a proportion of the facilities that are providing that service.

Thank you for this comment. We have now added two columns to Table 2 to report the proportions of health facilities that provide each service and had completed source documents’ data and HMIS data for the reporting period.

• Results, figure 1: Excellent figure. Consider adding a legend which reminds the reader which directions represent under/overreporting.

We have now added to Figure 1 a legend for over-reporting and under-reporting, as well as labels for data elements (see Fig 1).

• Table 4, “rare data elements”: I wouldn’t call these rare data elements as they are regularly reported into HMIS. Perhaps “rare events” or “rare outcomes”?

We have changed the name to “Data elements with rare events” (see Tables 1 and 4)

4) Discussion: Overall comment that the literature review should be updated to reflect where the current study fits in.

• Discussion, paragraph 2, 3rd sentence “These quality of Rwanda HMIS data… same data in other Sub-Sahara African countries”: The sentence notes “countries” but references only one study in one country.

Thank you for your comment. We have now rephrased the sentence and cited 3 studies in Ethiopia, Nigeria and Malawi. The text reads as follows: “The quality of Rwanda HMIS data on these data elements is higher than what was found in HMIS data verifications for the same data in Ethiopia40 and Nigeria41 and similar for the ANC1 indicator in Malawi22” (see page 21, lines 344-346).

• Discussion, paragraph 2, 6th sentence “This is a different finding to that found by other studies…by geographical location of reporting HFs”: Again, the sentence notes “other studies” but references only one study.

We have now cited two studies in South Africa and Ethiopia (see page 21, line 351).

• Discussion, paragraph 3: Please reflect more on the poor level of agreement, as there are notable directions in the level of agreements based on data element which are not elaborated – which indicators under/overreport and potential reasons. Discussion, paragraph 3, 2nd sentence “The accuracy of reporting on these data elements might be dependent on the health workers knowledge of how to calculate the pregnancy’s gestational age in weeks and how to correctly schedule the ANC standard visits, as well as the availability of tools, mainly pregnancy wheels, that facilitate these calculations.” While this may be true in terms of the validity of the documentation, the data verification exercise assesses the ability of the health care worker to report as expected based on the source documents. The external research team, whose data collection is being used as the reference standard for the recounted data, does not also measure the women for gestational age, schedule the appointments, etc. They are looking at the same data source, as the health facility staff would, for determining which numbers would be counted as ANC1_standard versus ANC new registrants.

Thank you so much for your comments. We have now clarified the link between the quality of HMIS data on these data elements and the knowledge of calculating pregnancy’s gestational age and availability of tools like pregnancy wheels that facilitate the calculations. The added text reads as follows: “It was observed by the study data collection team, that data in registers were recorded in ways that were not compatible with HMIS reporting, for example – gestational age at first visit recorded in the register in months and then reported into HMIS based on cutoffs in weeks which may contribute to challenges in reporting incompatible information out of the source documents into HMIS. In addition, an analysis of the ABC baseline data on the availability of essential medical equipment and supplies at facilities that received the ABC intervention, revealed that only 44.9% (31/69) of health centers were having a pregnancy wheel in the pre-intervention period” (see page 22, lines 362-369). We also highlighted in the “Methods/recounted data from facility source documents” that the recording of gestational age in units that were not compatible with the HMIS reporting on ANC1 and ANC4 standard visits data elements required the data collection team to recalculate the gestational age at each ANC visit. The text reads as follows: “In particular, due to observed variable unit of recording of gestational age (GA) in weeks or months in the ANC register by facility and care provider, ABC data collectors worked with midwives or nurses responsible for providing ANC to standardize the calculation of GA in weeks before recounting data on ANC1 and ANC4 standard visits, as the reporting to HMIS on these data elements is based on GA calculated in weeks. The data collection team used a pregnancy wheel and the data recorded on date of last menstrual period and dates of ANC visits for individual pregnant women who attended ANC to determine GA at each visit” (see page 10-11, lines 197-204). 

Reviewer #2’s comments: 

This a relevant paper as it addresses a crucial aspect of Health Management Information System (HMIS) data that is data quality assessment. HMIS are promising source data but data quality remains challenging in Sub-Saharan Africa countries including Rwanda. Unlike to most of the studies focusing on desk review of routine health information system data, this study verifies facility source documents and the level of agreement with national records.

Thank you.

1) However, it is a pity that the study is limited to only 76 districts within 4 districts that are non-representative of the entire Rwanda. In this is regard, it may be valuable to provide a brief context description of Rwanda heath system and health information system including the number of health facilities and districts in the country, the public and private sector, HMIS data collection process and data processing as that may impact the level of agreement.

Thank you. We have now added information on the number of districts and health facilities in Rwanda (see page 9, lines 157-159): “These MoH-operated facilities were included because they received the ABC intervention, and represented 14% (69/499) of health centers and 15% (7/48) of the hospitals in all 30 districts in Rwanda35”. We have also added detailed information on the HMIS data collection process under the methods section (see page 10, lines 180-190) which was also raised as a comment by Reviewer 1. 

2) Suggestion to revise the title that is a bit long and to make it more attractive (e.g. "Health Management Information System (HMIS) data verification: A case study in four districts in Rwanda"

The title now reads as follows: “Health management information system (HMIS) data verification: a case study in four districts in Rwanda”

3) In the abstract, good to clearly present the objective of the study, as this is not obvious, as well as improving the results and conclusion sections.

Thank you. We have now clearly stated the objective of our study in the introductory paragraph of the abstract and reads as follows: “We aimed to review the quality of Rwandan HMIS data for maternal and newborn health (MNH) based on consistency of HMIS reports with facility source documents” (see page 2, lines 26-28). 

We have also revised the abstract’s conclusion and reads as follows: “There was variable HMIS data quality by data element, with some indicators with high quality and also consistency in reporting trends across districts. Over-reporting was observed for ANC-related data requiring more complex calculations, i.e., knowledge of gestational age, scheduling to determine ANC standard visits, as well as quality indicators in ANC. Ongoing data quality assessments and training to address gaps could help improve HMIS data quality” (see page 3, lines 46-50).

4) My main comments are related to the objective of the study, analysis performed and the discussions of the results. 

• I agree that the verification of level of agreement between HMIS data and facility source documents data is a relevant objective. However, I do think that you may be able to go beyond this objective and tackle additional analysis. You may for instance carry out additional analysis like completeness of reporting and internal consistency for both HMIS data and facility source documents in order to highlight the impact of the lack of agreement between both sources on data quality. I wonder whether that is possible based on available data or at this stage, but it may be worthwhile to assess the impact of data accuracy (level of agreement) on data quality (e.g. completeness of reporting or internal consistency). A question can be to know whether districts with good level of agreement report data with better quality. Since you stated that (p.18) "this verification showed high level of agreement between data reported to HMIS and records in facility source documents for the number of ANC1, deliveries and live births", it may be interesting to assess data quality in districts with good agreement versus districts with low agreement.

Thank you. It would have been important to assess the effect of data completeness and consistency between related indicators on the level of agreement between HMIS data and records in facility source documents, however this is not possible for this study, as we only included in our analysis for HMIS data verification, facilities with complete HMIS and source documents for assessed data elements and the study period. In addition, we also have limitations to assess the consistency between related data elements as explained earlier in this reply letter that the consistency assessment with just a three-month reporting period would be subjected to wrong conclusions.

• To assess whether the source of data matter, it may be interesting to provide, even as an appendix, a table presenting the verification factor by source of data (ANC register, maternity register, PNC register, NCU register), and address a bit that in the analysis.

Thank you. We also think that the format of a source document (register) itself matters in reporting. However, we were not able to calculate a verification factor by source document, as one register was a source to multiple data elements that we assessed and it was not possible to aggregate them to do so. But, with the calculation of a verification factor by data element, we hope that this was indirectly captured. We have referenced in Table 1 the data source along with the description of each data element extracted. We have described the data source for each data element (see page 7, lines 142-147) and added a note in the methods (page 9, lines 172-173) when talking about the HMIS data sources to refer the reader to this table as follows: “Data are recorded using standardized registers developed by the MoH and provided to all HFs (see Table 1 for data sources)”.

5) Discussion:

• The discussion section needs to be deeply revisited to reflect more the expectations from a classical discussion section, discuss more the discrepancies between both sources, relevant factors (e.g. unmotivated, poor trained or overworked health personnel, disinterest for health data, etc., problem of equipment, potential issues regarding the transfer of data, data entry errors, etc.), implications of the results.

Thank you for your comment. We have included additional context in the discussion to better contextualize the results. We have added additional literature review that highlights some of the strengths of the data and compares our results to what has been seen in other countries, as raised by reviewer 1. In addition, we added a reference to a systematic review on data quality of immunization data, which identified key factors contributing to poor data quality. This revised paragraph on page 22, lines 356-359, now reads: “However, there was poor quality of HMIS data on the number of ANC1 and ANC4 standard visits with a general trend of over-reporting. A systematic review of immunization data quality identified insufficient human resources and limited healthcare worker capacity for reporting and using data as key issues that contribute to poor data quality23.”

• As an explication you stated (p. 18-19) "However, there was poor quality of HMIS data on the number of ANC1 and ANC4 standard visits. The accuracy of reporting on these data elements might be dependent on the health care provider’s knowledge of how to calculate the pregnancy’s gestational age in weeks and how to correctly schedule the ANC standard visits, as well as the availability of tools, mainly pregnancy wheels, that facilitate these calculations". Through this explanation, you are trying to address the reasons related to the true accuracy of the source documents, instead of providing relevant reasons explaining the lack of agreement between HMIS data and facility source documents. Suggestions to provide relevant reasons/explanations and in line with findings.

Thank you for your comment. We have now clarified the link between the quality of HMIS data on these data elements and the knowledge of calculating pregnancy’s gestational age and availability of tools like pregnancy wheels that facilitate the calculations. The added text reads as follows: “It was observed by the study data collection team, that data in registers were recorded in ways that were not compatible with HMIS reporting, for example – gestational age at first visit recorded in the register in months and then reported into HMIS based on cutoffs in weeks which may contribute to challenges in reporting incompatible information out of the source documents into HMIS. In addition, an analysis of the ABC baseline data on the availability of essential medical equipment and supplies at facilities that received the ABC intervention, revealed that only 44.9% (31/69) of health centers were having a pregnancy wheel in the pre-intervention period” (see page 22, lines 362-369). We also highlighted in the “Methods/recounted data from facility source documents” that the recording of gestational age in units that were not compatible with the HMIS reporting on ANC1 and ANC4 standard visits data elements required the data collection team to recalculate the gestational age at each ANC visit. The text reads as follows: “In particular, due to observed variable unit of recording of gestational age (GA) in weeks or months in the ANC register by facility and care provider, ABC data collectors worked with midwives or nurses responsible for providing ANC to standardize the calculation of GA in weeks before recounting data on ANC1 and ANC4 standard visits, as the reporting to HMIS on these data elements is based on GA calculated in weeks. The data collection team used a pregnancy wheel and the data recorded on date of last menstrual period and dates of ANC visits for individual pregnant women who attended ANC to determine GA at each visit” (see page 10-11, lines 197-204). 

6) Row 233; Check the median VF for syphilis as it does not with data in Table 2.

Thank you. We have now corrected the numbers for the median and interquartile range in the text (see page 16, lines 274-275).

7) Table 3: Write properly the label of data elements.

Thank you. We have now updated labels of data elements in all tables to avoid abbreviations (see Tables 1, 2, 3 and 4).

---

## [Decision Letter · Decision Letter 1]

24 Jun 2020

Health management information system (HMIS) data verification: a case study in four districts in Rwanda

PONE-D-19-35132R1

Dear Dr. Nshimyiryo,

We’re pleased to inform you that your manuscript has been judged scientifically suitable for publication and will be formally accepted for publication once it meets all outstanding technical requirements.

Kind regards,

Dejing Dou, Ph.D.

Academic Editor

PLOS ONE

Additional Editor Comments (optional):

Reviewers' comments:

Reviewer's Responses to Questions

**Comments to the Author**

1. If the authors have adequately addressed your comments raised in a previous round of review and you feel that this manuscript is now acceptable for publication, you may indicate that here to bypass the “Comments to the Author” section, enter your conflict of interest statement in the “Confidential to Editor” section, and submit your "Accept" recommendation.

Reviewer #2: All comments have been addressed

Reviewer #3: All comments have been addressed

2. Is the manuscript technically sound, and do the data support the conclusions?

Reviewer #2: Yes

Reviewer #3: Yes

3. Has the statistical analysis been performed appropriately and rigorously? 

Reviewer #2: Yes

Reviewer #3: Yes

4. Have the authors made all data underlying the findings in their manuscript fully available?

Reviewer #2: Yes

Reviewer #3: Yes

5. Is the manuscript presented in an intelligible fashion and written in standard English?

Reviewer #2: Yes

Reviewer #3: Yes

6. Review Comments to the Author

Reviewer #2: (No Response)

Reviewer #3: This paper addresses the data quality assessment problem of HMIS by studying the level of agreement between HMIS data and records in facility source documents. The revised version addresses the comments and is well written.

7. PLOS authors have the option to publish the peer review history of their article (what does this mean?). If published, this will include your full peer review and any attached files.

Reviewer #2: **Yes: **Abdoulaye Maïga, PhD

Reviewer #3: No

---

## [Editor Report · Acceptance letter]

6 Jul 2020

PONE-D-19-35132R1 

Health management information system (HMIS) data verification: a case study in four districts in Rwanda 

Dear Dr. Nshimyiryo:

I'm pleased to inform you that your manuscript has been deemed suitable for publication in PLOS ONE. Congratulations! Your manuscript is now with our production department. 

Kind regards, 

on behalf of

Professor Dejing Dou 

Academic Editor

PLOS ONE